# Anodic Aluminum Oxide-Based Chemi-Capacitive Sensor for Ethanol Gas

**DOI:** 10.3390/nano14010070

**Published:** 2023-12-26

**Authors:** Gi-Hwan Lim, In-Yea Kim, Ji-Young Park, Yong-Ho Choa, Jae-Hong Lim

**Affiliations:** 1Department of Materials Science and Engineering, Gachon University, Seongnam-si 13120, Republic of Korea; khl2ses@gachon.ac.kr (G.-H.L.); kiy7484@gachon.ac.kr (I.-Y.K.); 2Department of Materials Science and Chemical Engineering, Hanyang University, Ansan 15588, Republic of Korea; pjiyoung74@gmail.com (J.-Y.P.); choa15@hanyang.ac.kr (Y.-H.C.)

**Keywords:** sensor, AAO, capacitive sensor, surface modification

## Abstract

Alcohol ingested by humans can be analyzed via breath tests; however, approximately 1% can be excreted via the skin. In this paper, we present a capacitive sensor using hydrophobically treated anodic aluminum oxide (AAO) capable of detecting alcohol excreted through the epidermis. The degree of hydrophobicity based on the duration of exposure to 3-aminopropyltriethoxysilane vapor comprising a small number of Si–NH_2_ functional groups on the AAO surface was confirmed and the optimal exposure time was confirmed to be 60 min. The hydrophobized AAO showed a 4.8% reduction in sensitivity to moisture. Simultaneously, the sensitivity of the sensor to ethanol decreased by only 12%. Lastly, the fabricated sensor was successfully operated by attaching it to an ankle-type breathalyzer.

## 1. Introduction

Ethanol is a volatile gas that is closely associated with humans [1]. The detection and analysis of ethanol gas in human breath as well as the rapid and accurate detection of ethanol in the food and beverage industry is crucial [2,3]. Various protocols are used to determine the alcohol concentration inside the human body; the most widely used method involves measuring alcohol concentration in the breath. The measurement methods include the use of fuel cells, colorimeters, infrared absorption, dual sensors, and semiconductors, with the most commonly used one in Korea being the fuel-cell method that uses a platinum electrode [4,5,6]. This method detects the current generated by the chemical oxidation reaction that occurs when alcohol released through respiration attaches to the surface of a platinum electrode. Thus, this method can easily detect the presence and concentration of alcohol in breath. However, because these devices measure the content of a respiratory gas, widening their scope of application is difficult.

Given the growing concerns over crimes being committed after alcohol consumption, an increasing interest has been observed in implementing continuous monitoring of individuals who have committed serious offenses under the influence of alcohol, such as sex offenders. The aim of such monitoring is to track activities and prevent future crimes. However, relying on breath analysis to determine alcohol consumption poses practical difficulties owing to the significant amount of labor and time required. To calculate the alcohol concentration in the body, the amount of alcohol detected via breath analysis was calculated using a distribution ratio of 1:2100 [7]. However, measuring alcohol through the skin allows the detection of only ∼1% of the consumed alcohol [8]. The concentration of skin excretion per unit amount of alcohol ingested cannot be generalized because it depends on gender, weight, and metabolic activity [7]. A method for measuring ethanol via the skin should be capable of measuring sub-ppm levels because of the lower concentration of alcohol excreted via the skin than that in the breath. 

The method must be suited to the room-temperature application and should detect subtle changes in alcohol concentration. Therefore, capacitance-based sensors that do not involve charge movement and offer high reliability and excellent sensitivity are suitable. Capacitance-based gas sensors have a simple capacitor structure. These sensors have the advantage of reacting with molecules down to the level of a single molecular layer, ensuring high sensitivity and excellent response and recovery times [9,10,11]. Capacitance-based gas sensors have certain advantages over other types of sensors, such as low-temperature dependence, excellent thermal stability, linear response, and low power consumption [12,13,14]. In particular, a capacitive gas sensor was selected in this study, owing to its advantageous characteristics of possessing a simple structure, miniaturizability, and ability to allow continuous observation. A capacitive gas sensor typically comprises an upper electrode, a lower electrode, and a dielectric material [15]. Among the various dielectric materials available, mesoporous nanostructures, such as anodic aluminum oxide (AAO), have been widely used in chemical gas and vapor sensing applications. AAO has exceptionally reversible adsorption properties, which enable the selective adsorption of diverse gas molecules within its internal pores [16]. In addition, AAO is corrosion-resistant, thermally stable, and mechanically strong [16,17]. The sensing mechanism involves changes in the capacitance induced by variations in the dielectric constant of the dielectric material in the presence of the target gas. Specifically, the capacitance of the material changes when the target gas enters the pores of the sensor and interacts with the sensitive AAO material. Thus, monitoring this change in capacitance allows the detection of gases. 

However, because the permittivity of moisture is higher than that of ethanol, the sensitivity of the capacitive sensor to ethanol is affected by humidity. Thus, AAO materials used in sensors require surface treatment to prevent the adsorption of moist air. Vapor deposition techniques, such as plasma, atomic layer, and chemical vapor deposition, can be used as surface hydrophobization treatment methods; wet chemical techniques like raft polymerization and silanization can also be utilized [18,19]. Among these, wet silanization is a relatively easy method for effectively treating nanopores inside the AAO structure.

In this study, we developed a capacitive sensor utilizing silanization-treated AAO to minimize its sensitivity to humidity. We hypothesized that the silanization treatment would improve the surface characteristics of AAO and reduce its sensitivity to humidity, which was experimentally validated. The sensitivity of the sensor to ethanol was also investigated. The experimental results confirmed that the developed sensor reduced the effect of humidity by 34% in relation to existing sensors and increased the sensitivity to ethanol by 88%. These findings indicate the utility of silanization treatment for sensing performance enhancement. This study contributes to the improvement in the stability of sensors in humidity-sensitive environments and can lead to enhanced sensing performance in applications like ethanol detection.

## 2. Materials and Methods

### 2.1. Capacitive Gas Sensor Preparation

AAO (Whatman, Anodisc 13, Buckinghamshire, UK; diameter = 13 mm, thickness = 60 μm, pore size = 200 nm, and porosity = 30%) was used as a sensitive material for gas detection. An Au electrode was deposited on both sides of the AAO substrate using a direct current (DC) sputtering system. The thickness of the upper electrode of the sensor was adjusted experimentally, whereas that of the lower electrode was fixed at 100 nm. The electrode thickness of the no-pattern electrode was set at 50, 100, and 200 nm. To measure the capacitance of the sensor, copper wires were connected to the electrodes above and below the sensor using a silver paste.

### 2.2. Surface Modification

Before proceeding with surface treatment, the AAO was immersed in deionized (DI) water at 100 °C for 1 min to generate many hydroxyl groups, and then dried in an oven at 70 °C for 1 h and used. Surface treatment and silanization were performed using 3-aminopropyltriethoxysilane (APTES; 99%, Sigma-Aldrich, St. Louis, MO, USA). Silanization was performed thrice. The sensor was immersed in a 10 mM APTES toluene solution for 30, 60, and 120 min [20,21]. After the surface treatment, the sensors were rinsed with ethanol and DI water and dried under a stream of nitrogen gas. The sensor manufacturing process is shown in Figure 1.

### 2.3. Characteristics

The pore size and porosity of the AAO were observed using field-emission scanning electron microscopy (FE-SEM; HITACHI S-4300, Tokyo, Japan). In addition, the pore size of the sensor, in which the electrode was formed, was determined. Fourier-transform infrared (FT-IR) spectroscopy (Bruker, Vertex 70, Billerica, MA, USA) confirmed the presence or absence of hydroxyl groups that cause the adsorption of water and other functional groups [22]. Subsequently, the water contact angle (WCA; SEO, Phoenix-300, Suwon, Republic of Korea) was measured to verify the degree of hydrophobicity of the sessile droplets [23].

The capacitances of the devices were measured using an LCR meter (WAYNE KERR, 41100, Bognor Regis, UK) in the frequency range from 100 Hz to 1 MHz at ~20 °C upon exposure to analytes like dry air, wet air, and ethanol gas at different concentrations. The experimental protocol to determine the sensing mechanism was initiated by exposing the sensors to dry air to reach a steady-state (equilibrium) baseline, followed by exposure to the analyte at a fixed concentration for 1 min and measuring the capacitance of the sensors. Once the measurements were completed, the processes were repeated for a higher concentration. To minimize the change in concentration due to the accumulation of gas in the chamber, a stabilization time of 1 min was allowed between the concentration changes. To eliminate the effect of temperature on the converter, the measured temperature was set at ~20 °C. Ethanol was delivered to the sensor at flow rates of 2.5, 5, 10, 20, 30, 40, and 50 ppm for 1 min each, and dry air was delivered as a pulse. The effect of humidity, such as the sensor response to ethanol, was determined by injecting wet air instead of ethanol at the same flow rate. Figure 2 shows a schematic of the settings used to measure the sensor. The gas flow into the gas chamber was controlled through three mass flow controllers (MFCs). For example, to measure the ethanol concentration of 50 ppm, the ethanol flow rate was set to 50 sccm and the dry air flow rate to 450 sccm. Table 1 shows the amount of gas injected for each ethanol concentration. The humidity concentration was measured over the range of 4.5–90% relative humidity (RH) at measured ethanol gas concentrations of 2.5, 5, 10, 20, 40, and 50 ppm. Dry air was used as the base gas and ethanol gas and wet air were used as the target gas. The total flow values of the base and target gases were 50 sccm. Ethanol gas was used at a concentration of 500 ppm and flowed along with dry air. The humidity in the chamber was maintained at approximately 90% RH by injecting dry air into a water bubbler to be mixed with dry air. The air that passes through the water bubbler is called wet air. Table 2 shows the flow rates of wet and dry air for each humidity. 

The sensitivity of a sensor is important because it indicates its detection performance. Capacitance (*C*) can be obtained simply by dividing the dielectric constant (*ε*) of the dielectric materials and the electrode area (*A*) (Equation (1)) [24].
(1)$$C=\frac{\varepsilon A}{d},$$

Figure 3 shows the AAO membrane in contact with one pore of the sensor as an equivalent circuit. The sensor’s detailed capacitors include capacitors for AAO ($C_{\mathrm{AAO}}$), air ($C_{\mathrm{Air}}$), and target gas ($C_{\mathrm{Gas}}$). The capacitor of the entire sensor will be referred to as $C_{\mathrm{Total}}$ (Equation (2)).
(2)$$C_{\mathrm{Total}}=C_{\mathrm{AAO}}+C_{\mathrm{Air}}+C_{\mathrm{Gas}},$$

In this study, the sensitivity (*S*) of the sensor was calculated by dividing the capacitance change (Δ*C*) obtained when the target gas is injected by the capacitance ($C_{0}$) obtained when only dry air is injected (Equation (3)) [25].
(3)$$S=\frac{\Delta C}{C_{0}}=\frac{C_{\mathrm{Total}}-C_{0}}{C_{0}},$$

The sensitivity of the subsequent sensor is expressed as the calculated value of the measured capacitance.

## 3. Results and Discussion

### 3.1. Improving the Sensitivity of the Sensor

#### 3.1.1. Change in Thickness of Sensor Electrode

Figure 4 shows the SEM images of the AAO surfaces used as the sensing material for different Au electrode thicknesses. Figure 4a shows the AAO surface without electrode deposition, indicating that the pore size of pure AAO is ~200 nm. Figure 4b–d show SEM images of the surface of the sensor with electrode thicknesses of 50, 100, and 200 nm, respectively, with the corresponding open pore sizes being approximately 120–150 nm, 110–170 nm, and 80–120 nm. The open-area ratios shown in Figure 4a–d are approximately 75%, 67.5%, 60%, and 30%, respectively. These SEM images confirmed that, as the thickness of the electrode increased, the Au particles deposited by sputtering blocked the pores of AAO, thus decreasing the pore size [26]. The sensitivity of the sensor was analyzed for different electrode thicknesses to determine the optimal Au thickness. 

Figure 5a–c show the sensitivity of the sensor to ethanol for electrode thicknesses of 50, 100, and 200 nm. 

The sensitivity of the sensor increased with increasing ethanol concentration. At a gas concentration of 30 ppm, the sensitivities were 0.07, 0.1, and 0.065 for the electrode thicknesses of 50, 100, and 200 nm, respectively. Thus, the sensitivity increased as the electrode thickness increased from 50 to 100 nm. However, the sensitivity decreased when the electrode thickness was 200 nm. The sensitivity increased as the electrode thickness increased from 50 to 100 nm because the capacitance increased with increasing electrode area, as shown in Equation (1). The sensitivity decreased at an electrode thickness of 200 nm because an increase in electrode thickness beyond 100 nm inhibits the diffusion of ethanol gas into the AAO by acting on the pores of the material, as confirmed by the SEM images shown in Figure 4. Figure 5d presents a graph depicting the change in sensitivity with respect to alcohol concentration for different electrode thicknesses. The sensitivity of the 100 nm electrode was markedly higher than those of the 50 and 200 nm electrodes. Figure 5e shows a graph of the response/recovery times of a sensor fabricated using 100 nm electrodes. The response and recovery times indicate the time at which 90% of the change in capacitance caused by the inflow of the target gas is achieved. It was confirmed that the ethanol gas concentration has no significant effect on the response time.

#### 3.1.2. Change in the Measuring Frequency of the Sensor

Dielectric materials exhibit frequency-dependent properties owing to the time constants associated with their electrical dipoles. To analyze the performance of the fabricated AAO gas sensors at different frequencies, we determined the change in sensor response in the presence of ethanol at different frequencies at gas concentrations in the range of 2.5–50 ppm. Measurements were performed after sensor stabilization. 

Figure 6a–d depict graphs showing the sensitivity of the sensor at measurement frequencies of 0.5, 1, 10, and 100 kHz, respectively. 

When the measurement frequency was 1 kHz (Figure 6b), the sensor sensitivities to ethanol were 0.0075, 0.020, 0.029, 0.042, 0.053, 0.061, and 0.068. The sensitivity of the sensor increased with measurement frequency (Figure 6). When the concentration of ethanol gas was increased from 2.5 to 5 ppm, the slopes of the sensor sensitivity changes were 0.75, 0.85, 0.56, and 0.49 at measurement frequencies of 0.5, 1, 10, and 100 kHz, respectively. This is because the permittivity of the dielectric material changes with frequency. As the dipole moment of the material increases, its frequency decreases. When the dipole moment increases, the polarization increases. Therefore, as the frequency increases, owing to polarization, it is difficult to follow the electric field, and the permittivity decreases. [27,28]. The sensitivity of the sensor was higher because the polarity and the adsorption of molecules are both higher at lower frequencies. The signal-to-noise ratio (SNR) when the measurement frequency is 0.5 kHz is 22.27 times greater than that when the measurement frequency is 1 kHz; additionally, the noise decreases as the measurement frequency increases. Although the sensitivity of the sensor was highest when the measurement frequency was 0.5 kHz, subsequent measurements were performed at 1 kHz due to low SNR [29]. Figure 6f shows a response/recovery time graph of the optimized sensor measured at a frequency of 1 kHz. It was confirmed that the ethanol gas concentration has no significant effect on the response/recovery time.

The thus-fabricated sensor is a capacitive sensor made of metal–oxide–metal spheres. Figure 3 shows a schematic of the equivalent circuit of the fabricated sensor, which indicates the relationship between one pore and the AAO. Therefore, there are expected to be innumerable equivalent circuits of the entire sensor. 

### 3.2. Decreased Moisture Sensitivity of the Sensor 

Capacitive gas sensors are used to detect changes in the dielectric constant upon adsorption of the target gas onto the sensing material of the sensor. The dielectric constants of dry air, alcohol vapor, and water vapor, which can affect alcohol sensor measurements, are 1.0006, 1.0061, and 1.0126, respectively [30]. Therefore, capacitive gas sensors are inherently susceptible to moisture [31]. With increasing dielectric constant, the sensitivity of the sensor increases because the polarization shape is easily generated by the electric field. In addition, because the measured capacity changes according to the applied frequency, sensitivity analysis must be performed at a set frequency. The sensor measurement conditions and the thickness of the electrode were both measured using a 100 nm electrode with a measurement frequency of 1 kHz. The dielectric constant of moisture is higher than that of ethanol gas. Therefore, moisture must be prevented from being absorbed into the sensor. Thus, various studies were conducted to suppress the susceptibility of the sensor to moisture.

Figure 7 shows the change in sensitivity of the capacitive sensor according to humidity concentrations. 

As shown in Figure 7a, the sensitivity increased to 0.031, 0.032, 0.031, 0.042, 0.95, 4.29, and, finally, 24.15 as the RH increased to 90%. This result is consistent with the aforementioned capacitance results. The dielectric constant of the insulator increases with the concentration of water molecules adsorbed on the AAO surface. Therefore, to exclude the effect of moisture, preventing moisture adsorption by hydrophobic treatment on the surface of AAO, which is an insulator, is necessary. Figure 7b shows the sensor response/recovery time as a function of humidity.

Figure 8 shows the FT-IR spectra of the AAO surface according to the APTES exposure time during the hydrophobic treatment of AAO. 

Table 2 was attached to the hydroxyl group present on the surface using APTES [32]. In addition, the hydrophobicity increases by generating excess OH groups on the surface by steam treatment. Therefore, the O–H stretch band in the FT-IR spectra was used to determine the number of remaining hydroxyl groups. The peak at 1133 cm−^1^ is assigned to the Si–O–Si linkage [33]. The peak at approximately 1698 cm−^1^ is due to the vibration of hydroxyl groups on the AAO surface [34]. The bands at 2853 and 2928 cm−^1^ are attributed to the valence stretching vibration of C–H bonds [35]. Lastly, the peaks at approximately 3610 and 3740 cm−^1^ are attributed to the O–H and N–H stretching bands, respectively [36]. In our study, N–H and Si–O–Si bands were formed on the AAO surface after APTES treatment. Figure 9 shows the hydroxyl groups on the AAO surface.

After silanization, a significant portion of the hydroxyl groups was removed and combined with silane to form Si–NH_2_ [25]. Therefore, successful silanization was confirmed on the AAO surface by FT-IR spectroscopy. The hydroxyl group peak shown in the FT-IR spectra is attributed to the remaining hydroxyl groups on the surface after processing to attach functional groups before the surface treatment. Therefore, additional measurements were performed to confirm the success of surface treatment. The hydrophobicity of the surface and sensor was confirmed by measuring the WCA using the sessile drop method [37].

The WCA confirmed the hydrophobicity according to the silanization of the AAO surface. As shown in Figure 10a, the WCA analysis of pure AAO without surface treatment confirmed that the surface angle was extremely small at 12.6°, confirming that AAO was hydrophilic. Figure 10b–d show that the WCA of the samples surface-treated with AAO exposed to APTES 10 mM vapor for 30, 60, and 120 min were determined to be 96.37°, 112.1°, and 110.1°, respectively. This contact angle used the average of the left and right angles indicated by the red lines in the figure. Thus, the AAO surface became hydrophobic when exposed to APTES and became more hydrophobic with increasing time. Surface hydrophobization was successfully performed using covalent bonds between the hydroxyl groups and silanes on the AAO surface [38,39].

The sensitivity of the sensor was determined using the principle of change in permittivity due to the covalent bonds between AAO and various gases. Water has a slightly higher dielectric constant than ethanol. Therefore, to reduce the sensitivity of the sensor to moisture, a surface treatment was performed on the sensing material. The sensor underwent surface treatment with APTES for 30, 60, and 120 min. The change in capacitance with humidity was measured using the treated sensor. Figure 11a shows the sensitivity response of the sensor without surface treatment and Figure 11b–d show those of the sensors that have undergone surface treatment for 30, 60, and 120 min, respectively. 

The sensitivities of the sensors at 54% RH for surface treatment times of 0, 30, 60, and 120 min were 2.1, 0.19, 0.05, and 0.07, respectively. In addition, the sensitivity of the sensor slowly increased below 45% RH, during which the adsorption of water molecules and chemicals occurred. The sensitivity of the sensor increased rapidly beyond 45% RH, at which point physical adsorption occurred. The hydroxyl groups of AAO and water molecules underwent chemical adsorption and, when the RH exceeded 45%, physical adsorption occurred.

Therefore, with increasing humidity concentration, the sensitivity of the sensor increased rapidly as physical adsorption proceeded. This effect was confirmed by the slope of the sensitivity-to-moisture curve of the sensor before and after 45% RH. 

The gas selectivity of the sensor was improved by reducing the chemical adsorption by surface treatment. Figure 12a shows the results of measuring ethanol gas concentration using a sensor fabricated with a surface-treated dielectric material. The treated sensor composed of the material that underwent silanization by 10 mM APTES for 60 min showed a sensitivity similar to that of the bare sensor. The response and recovery times of the sensor surface treated for 60 min at an ethanol concentration of 10 ppm were measured to be 27.54 and 26.26 s, respectively, confirming that the response and recovery times were similar to those of the bare sensor. However, the sensitivity of the sensor to ethanol decreased for those sensors composed of dielectric material that had undergone 30 and 120 min of silanization. Figure 12b shows a graph summarizing the moisture sensitivity of the surface-treated sensor. The sensor surface-treated using 10 mM APTES for 60 min exhibited a reduced sensitivity to ethanol of ~12% and a reduced sensitivity to moisture of ~4.8% compared with that of the bare sensor at 54% RH. The sensitivity of the sensor composed of the treated dielectric material was maintained at 45% without interference from water vapor. Table 3 shows a comparison of the characteristics of capacitive gas sensors using various sensing materials. The results demonstrate that the sensor using AAO exhibited short response/recovery time at low ethanol concentrations.

### 3.3. Sensing Performance of Capacitive Type Sensor Device

Figure 13a shows a photograph of an individual wearing an electronic device with an embedded AAO sensor on the ankle. 

Figure 13b shows the capacitance of the AAO sensor without surface treatment; the capacitance was measured by attaching the device after the individual had begun drinking. The red arrow in Figure 13 indicates the time when drinking began, and the red arrow indicates the time during which drinking continued. The subject imbibed an alcoholic drink from 960 to 1260 s and consumed 360 mL of 16.9% alcohol. When using the AAO sensor that had not undergone surface treatment, the change in capacitance according to the degree of drinking could not be measured because of factors like humidity and temperature. However, the hydrophobic AAO sensor confirmed the capacitor response to drinking. As shown in Figure 13c, the subject drank 360 mL of 16.9% alcohol from 480 to 1680 s. After measurement, the sensitivity of the sensor increased from 0.025 to 0.04. After 7500 s, the sensitivity was maintained at a higher level than that initially measured. Thus, the synthesized AAO can function as an alcohol sensor. Although the sensor’s sensitivity to moisture has been reduced by surface treatment, it still shows high sensitivity. Therefore, interference with moisture should be further reduced in future studies. It should also be noted that sensitivity to ethanol gas decreases after surface treatment.

## 4. Conclusions

In this study, a capacitive alcohol sensor with controlled moisture sensitivity was developed using AAO to detect alcohol in the human epidermis. The sensitivity was determined according to the frequency of the manufactured AAO, and the frequency affording the highest sensitivity was confirmed to be 1 kHz. In addition, hydrophobization was performed by exposure to APTES to reduce sensitivity to moisture; the sample exposed for 60 min to 10 mM APTES exhibited optimal hydrophobization. An ankle-type alcohol sensor device fabricated from hydrophobic AAO could be used to monitor changes after alcohol had been imbibed, and changes in sensitivity according to the influx of alcohol into the human body were confirmed. Thus, the hydrophobic AAO can be used as a capacitive alcohol sensor. In particular, the sensor allowed continuous detection by stably detecting the ethanol gas at the ppm level. This successfully decreases the number of variables by reducing the interference by moisture. However, after surface treatment, the sensitivity of the sensor decreased slightly. In addition, the sensor still has the disadvantage of interference from moisture. In future, we will conduct research to increase sensitivity to the target gas and lower sensitivity to moisture in the sensor detection area.

## Figures and Tables

**Figure 1 nanomaterials-14-00070-f001:**
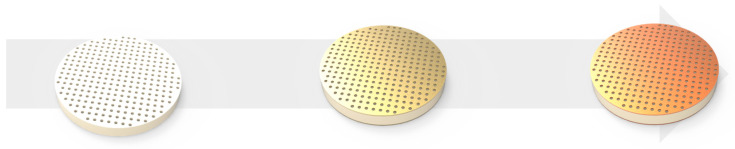
Schematic of the anodic aluminum oxide (AAO) sensor manufacturing process of AAO formation, Au sputtering, and surface modification, sequentially.

**Figure 2 nanomaterials-14-00070-f002:**
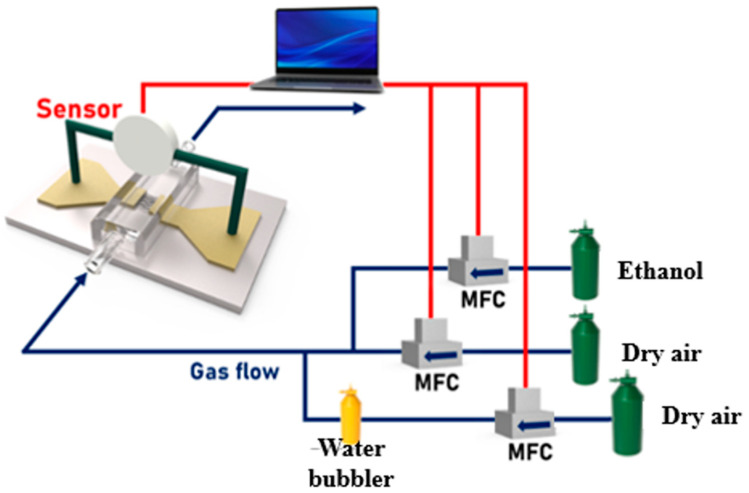
Schematic of the gas-sensing system of the capacitive sensor using AAO.

**Figure 3 nanomaterials-14-00070-f003:**
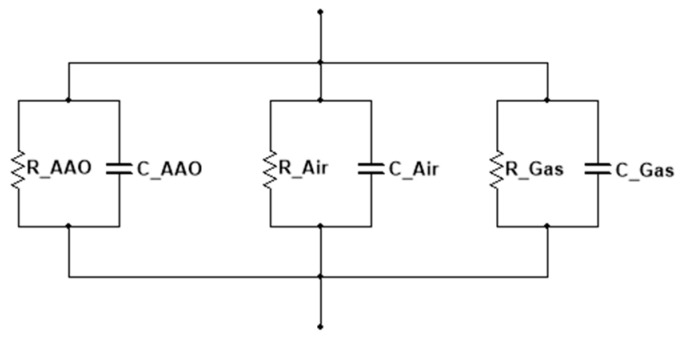
Equivalent circuit of the AAO sensor.

**Figure 4 nanomaterials-14-00070-f004:**
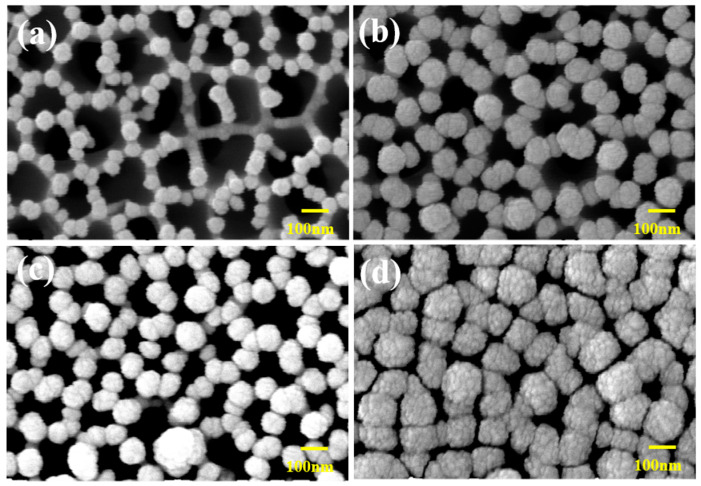
Scanning electron microscopy images of AAO surface for different Au deposition thicknesses: (**a**) bare AAO, (**b**) 50 nm Au electrode, (**c**) 100 nm Au electrode, and (**d**) 200 nm Au electrode.

**Figure 5 nanomaterials-14-00070-f005:**
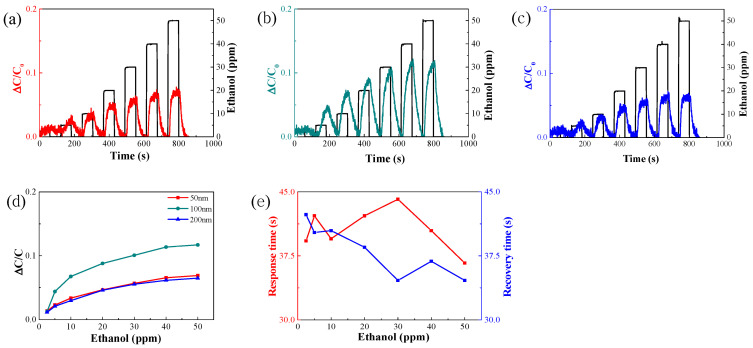
Sensitivity responses of sensors with electrode thicknesses of (**a**) 50, (**b**) 100, and (**c**) 200 nm. (**d**) Change in sensor sensitivity as a function of electrode thickness. (**e**) Response/recovery times of the sensor using 100 nm-thick electrodes.

**Figure 6 nanomaterials-14-00070-f006:**
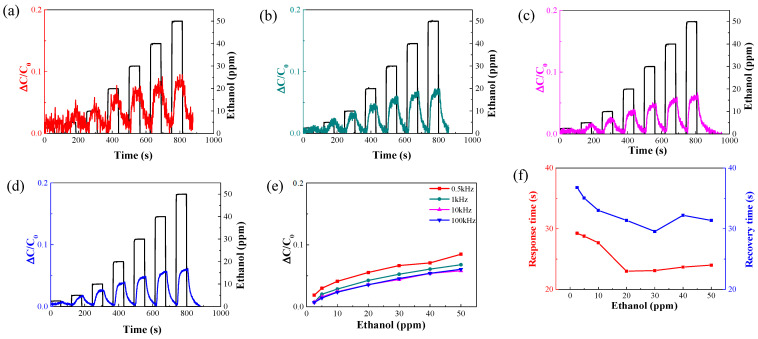
Response of sensor sensitivity at frequencies of (**a**) 0.5, (**b**) 1, (**c**) 10, and (**d**) 100 kHz. (**e**) Sensor sensitivity changes according to frequency. (**f**) Response/recovery times of the sensor after setting the measurement frequency to 1 kHz.

**Figure 7 nanomaterials-14-00070-f007:**
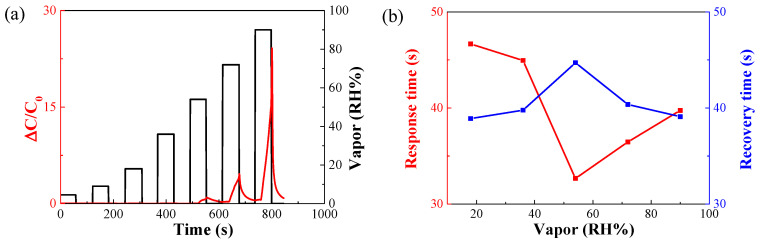
(**a**) Response of sensor sensitivity at different humidity concentrations at 1 kHz frequency. (**b**) Response/recovery times in response to sensor humidity.

**Figure 8 nanomaterials-14-00070-f008:**
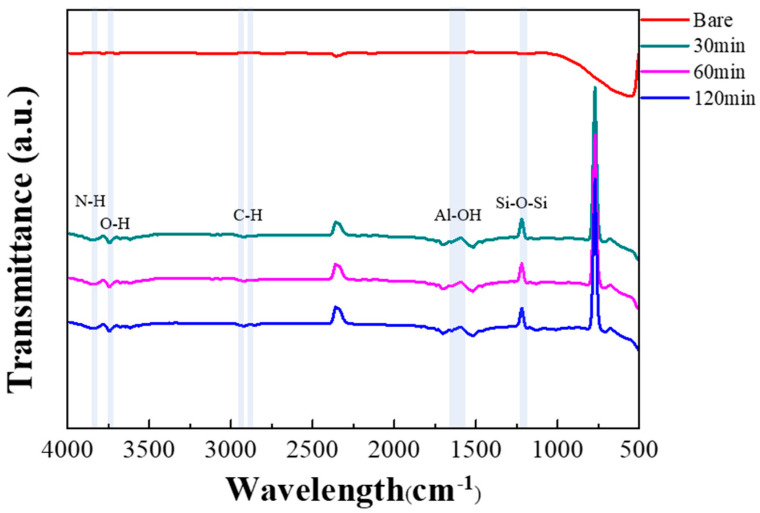
Fourier-transform infrared spectra for different durations of silanization treatment of AAO surface by 3-aminopropyltriethoxysilane.

**Figure 9 nanomaterials-14-00070-f009:**
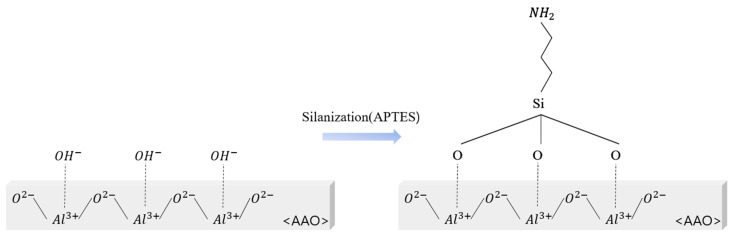
Schematic of silanization for the hydrophobization of AAO.

**Figure 10 nanomaterials-14-00070-f010:**
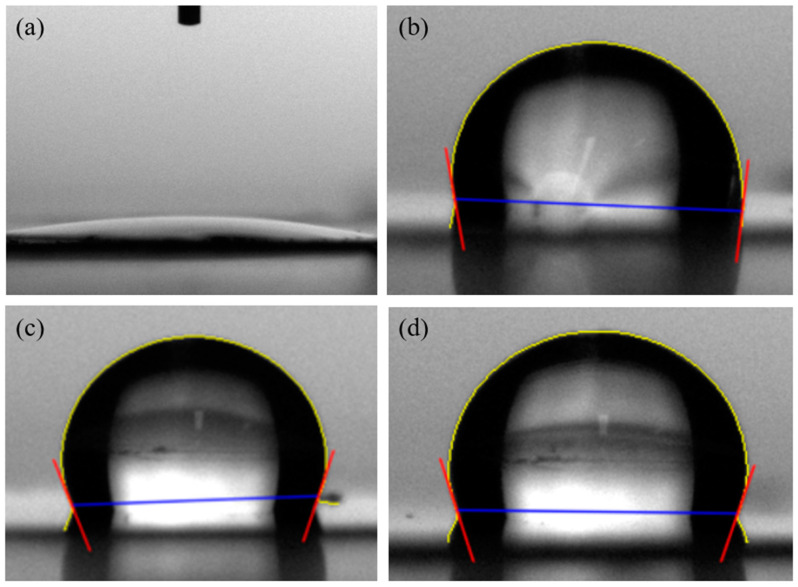
Water contact angles for different durations of silanization treatment: (**a**) 0 (untreated), (**b**) 30, (**c**) 60, and (**d**) 120 min.

**Figure 11 nanomaterials-14-00070-f011:**
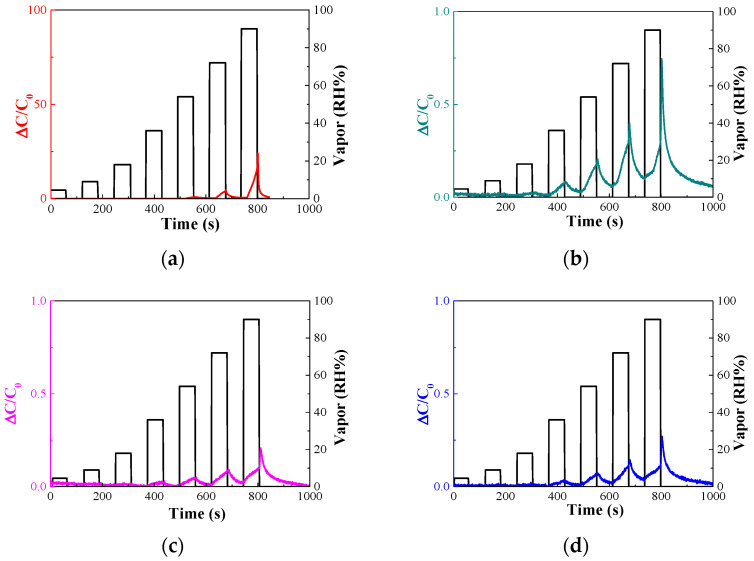
Sensitivity responses of sensors that have been surface-treated with 10 mM APTES for (**a**) 0, (**b**) 30, (**c**) 60, and (**d**) 120 min.

**Figure 12 nanomaterials-14-00070-f012:**
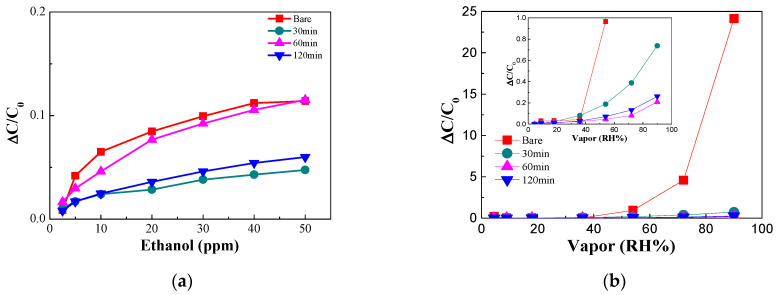
Sensor sensitivity characteristics of (**a**) ethanol gas and (**b**) water vapor obtained using different silanization-treated AAOs with the magnified inset.

**Figure 13 nanomaterials-14-00070-f013:**
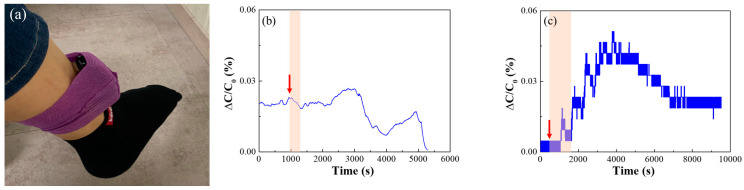
(**a**) Photograph of the ankle-wearing electronic alcohol sensor fabricated using the manufactured device. Results were obtained using (**b**) a sensor with an untreated surface and (**c**) a sensor with its surface treated with 10 mM APTES. (Red arrow; the time when drinking began, colored area; time during which drinking continued).

**Table 1 nanomaterials-14-00070-t001:** Flow rate of each injected gas according to the ethanol concentration.

Concentration [ppm]	Ethanol [sccm]	Dry Air [sccm]
2.5	2.5	497.5
5	5	495
10	10	490
20	20	480
30	30	470
40	40	460
50	50	450

**Table 2 nanomaterials-14-00070-t002:** Humidity control through a mixture of wet air and dry air.

Concentration [RH]	Wet Air [sccm]	Dry Air [sccm]
4.5%	25	475
9%	50	450
18%	100	400
36%	200	300
54%	300	200
72%	400	100
90%	500	0

**Table 3 nanomaterials-14-00070-t003:** Sensing performance of the developed and other capacitive sensors (*C*: concentration, Res/Rec time: response/recovery time, *T*: temperature).

Sensing Material	*C* (ppm)	Response	Res/Rec Time (s)	*T* (℃)	Ref.
Cobalt ferrite	7	1.4	180/180	31	[40]
Cu-BTC nanoporous	500	7.5	about 600/180	25	[41]
Cu-BTC thin film	250	48.6	about 140/140	25	[42]
AAO	10	0.05	27.54/26.26	25	this work

## Data Availability

Data underlying the results presented in this paper are not publicly available at this time but may be obtained from the authors upon reasonable request.

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
