# Peer review of "Anodic Aluminum Oxide-Based Chemi-Capacitive Sensor for Ethanol Gas"

_nanomaterials, 2023, doi:10.3390/nano14010070_

Round 1

Reviewer 1 Report

Comments and Suggestions for Authors

It should be mentioned if there is in the literature about the concentration of ethanol on the skin, depending on the amount of alcohol consumed..

Adding some information regarding the selectivity of the sensor to ethanol.

page 12 rows 363-364 "After measurement, the sensitivity of 363
the sensor increased from 0.25 to 0.04. " 0.025  (not 0.25).

Author Response

Dear reviewer

We thank you and the reviewers for your thoughtful suggestions and insights, which have enriched the manuscript and produced a better and more balanced account of the research. We hope that the revised manuscript is now suitable for publication in your journal.

Thank you for your consideration. I look forward to hearing from you.

Sincerely,

Jae-Hong Lim

Department of Materials Science and Engineering, Gachon University

Gyeonggi-do 13120, Republic of Korea

Email address: limjh@gachon.ac.kr

Reviewer 2 Report

Comments and Suggestions for Authors

In this paper (nanomaterials-2733296), the authors reported a capacitive ethanol sensor based on silanization-treated AAO. The method and gas sensing performance are basically acceptable. However, there are some problems in the motivations, experiment, data results, discussion and analysis. The manuscript may be accepted after major modifications.

1. Introduction: Semiconductor ethanol gas sensors seem to be more widely used, while the development of capacitive gas sensors seems to be slow. What are the advantages of capacitive gas sensors?

2. Figure 2: Please provide specific flow configurations for different ethanol concentrations and RHs. For example, creating a table.

3. The response and sensitivity of gas sensor cannot be confused, as sensitivity is the slope of the fitting equation for the linear response interval. Equation (3) should be expressed as “Response”, please refer to right supporting literature such as Sens. Actuators B Chem. 2022, 369, 132302. Check the relevant statements in the entire manuscript and figures.

4. It is recommended to test the effect of different RHs on ethanol response.

5. Response/recovery times need to be provided.

6. Is there any advantage in gas sensing performance? Please give the evaluation and compare with the reported ethanol sensors.

Comments on the Quality of English Language

 Minor editing of English language required

Author Response

Dear reviewer,

We thank you and the reviewers for your thoughtful suggestions and insights, which have enriched the manuscript and produced a better and more balanced account of the research. We hope that the revised manuscript is now suitable for publication in your journal.

Thank you for your consideration. I look forward to hearing from you.

Sincerely,

Jae-Hong Lim

Department of Materials Science and Engineering, Gachon University

Gyeonggi-do 13120, Republic of Korea

Email address: limjh@gachon.ac.kr

Round 2

Reviewer 2 Report

Comments and Suggestions for Authors

The revised manuscript looks better, and the authors have responded positively to some of the questions. However, some issues have not been substantially addressed.

1.       Response to question 1: After examination, reference [12] cannot support these viewpoints. Furthermore, these so-called advantages have not been reflected in this work.

2.       Response to question 2: It is necessary to provide flow values (such as sccm) for different pipeline flow meters, not just concentration results.

3.       Figure 3 is unclear, it is recommended to redraw it.

4.       “…sensitivities were 0.07, 0.1, and 0.065 for…”. The response and sensitivity of gas sensor cannot be confused, as sensitivity is the slope of the fitting equation for the linear response interval, please refer to right supporting literature such as Sens. Actuators B Chem. 2022, 369, 132302.

5.       Some unsupplemented experiments and existing shortcomings need to be explained in the revised manuscript, for example, the influence of humidity.

6.       Most of the references are outdated. In addition, the reference format does not meet the requirements of the journal.

7.       Check English writing.

8.       The resolution of the images seems insufficient.

Comments on the Quality of English Language

Minor editing of English language required.

Author Response

Dear Editor:

I wish to re-submit the manuscript titled “Anodic-Aluminum-Oxide-Based Chemi-Capacitive Sensor for Ethanol Gas.” The manuscript ID is nanomaterials-2733296.

We thank you and the reviewers for your thoughtful suggestions and insights. The manuscript has benefited from these insightful suggestions. I look forward to working with you and the reviewers to move this manuscript closer to publication in the Nanomaterials.

The manuscript has been rechecked and the necessary changes have been made in accordance with the reviewers’ suggestions. The responses to all comments have been prepared and attached herewith.

Thank you for your consideration. I look forward to hearing from you.

Sincerely,

Jae-Hong Lim
